# Naturido alleviates amyloid β$_{1-42}$-induced adverse effects in a transgenic *Caenorhabditis elegans* model of Alzheimer's disease

**Piyamas Sillapakong**[1☉]**, Tokumitsu Wakabayashi**[2☉]*****, Koichi Suzuki**[1†]

**1** Biococoon Laboratories, Inc., Morioka, Iwate, Japan, **2** Department of Chemistry and Biological Sciences, Faculty of Science and Engineering, Iwate University, Morioka, Iwate, Japan

† Deceased
☉ These authors contributed equally to this study.
* wakat@iwate-u.ac.jp

## Abstract

Alzheimer's disease (AD) is a progressive neurodegenerative disorder primarily associated with aging. While the amyloid hypothesis is not the only explanation for AD pathogenesis, it is widely recognized that the accumulation of amyloid β (Aβ) protein triggers pathological changes in the brains of patients. In a previous study, we showed that Naturido, a cyclic peptide derived from the medicinal fungus (*Isaria japonica*) grown on domestic silkworms (*Bombyx mori*), could reverse several age-related deficits in senescence-accelerated mice. In this study, we explored the potential of Naturido to reduce Aβ-related toxicity in transgenic *Caenorhabditis elegans* models of AD, where human Aβ$_{1-42}$ protein is overexpressed in neurons. Our results demonstrated that Naturido administration alleviated various phenotypes, including Aβ-induced impairment in associative learning, serotonin hypersensitivity, and locomotion in the transgenic *C. elegans*. These findings suggest the potential of Naturido as a candidate molecule for the prevention and/or treatment of AD.

## Introduction

Alzheimer's disease (AD) is a progressive neurodegenerative disorder associated with cognitive decline and irreversible memory loss. AD accounts for 60–80% of dementia cases, making it significant public health issue, particularly in countries with large aging population [1]. However, the number of available drugs for AD treatment is limited, highlighting the urgent needs for the development of safe and effective therapeutic options.

Although the exact cause of the pathological changes in AD remains controversial, the "amyloid cascade hypothesis", in which fibril formation of amyloid β (Aβ) protein and its accumulation leads to toxicity, is the most widely accepted pathological mechanism. Consequently, many newly developed treatments for AD treatment focus on reducing Aβ levels and inhibiting amyloid fibrils formation [1].

Along with the pharmaceutical drugs, natural compounds with neuroprotective effects have gained attention for their potential to reduce Aβ-induced toxicity [2]. Among these,

**Data availability statement:** All relevant data are included within the paper.

**Funding:** DKS Co., Ltd. (the parent company of Biococoon Laboratories, Inc.) provided support for this study in the form of salaries for PS and KS. The funders had no role in the study design, data collection and analysis, decision to publish, or preparation of the manuscript.

**Competing interests:** The authors have reviewed the journal's policy and disclose the following competing interests: PS and KS are paid employees of DKS Co., Ltd. (the parent company of Biococoon Laboratories, Inc.). This does not affect our adherence to PLOS ONE policies on sharing data and materials. There are no products in development or marketing to declare.

several plant-based and animal-based ingredients, such as Ginkgo extract (Egb761 and Ginkgolides [3]), plant polyphenols (quercetin and resveratrol [4,5]), buckwheat trypsin inhibitor protein (rBTI [6]), and bovine $\alpha_{s1}$-Casein [7], have been evaluated using transgenic *Caenorhabditis elegans* strains, that overexpress human $A\beta_{1-42}$ protein in muscle or neurons, serving as an AD model. In these studies, the natural ingredients have been shown to reduce oligomeric Aβ protein or overall Aβ protein levels, leading to the alleviation of Aβ-induced adverse effects in the transgenic worm.

Naturido, is a cyclic peptide derived from the medicinal fungus (*Isaria japonica*), grown on domestic silkworm (*Bombyx mori*). Our previous study demonstrated that upon the addition of Naturido to the culture medium, the peptide enhanced proliferation, exhibited anti-inflammatory effects, and promoted axonal and dendritic growth in primary cultured astrocytes, microglia, and hippocampal neurons, respectively [8]. Furthermore, oral administration of Naturido improved age-related learning and memory impairments in senescence-accelerated mice [8,9]. In this study, we evaluated the potential of Naturido to alleviate Aβ-induced toxicity in a transgenic *C. elegans* AD model. Naturido effectively alleviated several neuronal defects, caused by human Aβ protein, including associative learning.

## Materials and Methods

### Strains and culture

The transgenic *C. elegans* strains used in this study were CL2355, which express human amyloid β (1-42) protein ($A\beta_{1-42}$) upon a temperature upshift from 16°C to 23°C in neuronal cells, and CL2122, a control transgenic strain that does not express $A\beta_{1-42}$ [3,10]. All worms were maintained under standard conditions at 16°C [11], unless otherwise noted.

For the assays, transgenic worms were cultured on Nematode Growth Medium (NGM) plates, seeded with *E. coli* (OP50) as a food source. The culture plates were supplemented with various concentrations of Naturido or other reagents (as described below). To avoid metabolic alteration of these chemicals by bacterial cells, NGM plates with or without these chemicals were UV-irradiated for 20 min (254 nm, 5.0 J/cm² min by DNA-FIX, ATTO, Tokyo, Japan) to kill OP50 before use (UV-NGM). To obtain synchronously grown hermaphrodite worm populations free from bacterial contamination, eggs were prepared from gravid adult worms using the alkaline-bleach method [12]. The collected eggs were resuspended in a small volume of S-basal buffer, and approximately 200 eggs were plated on UV-NGM. The hatched worms were cultured at 16°C until the temperature upshift to 23°C to induce $A\beta_{1-42}$ expression.

### Reagents

Naturido, a cyclic peptide that reverses several age-related deficits in senescence-accelerated mice, was purified from the medicinal fungus (*Isaria japonica*) grown on domestic silkworms (*Bombyx mori*), as we described previously [8]. Donepezil hydrochloride (D6821, Sigma-Aldrich, St louis MO, USA) was used as a positive control reagent. A 50 µL drop of solution, containing various concentrations of Naturido or Donepezil hydrochloride, was added directly to the seeded UV-NGM and spread evenly across the plate. Final working concentrations are indicated in all figures. Serotonin creatinine sulfate salt (H7752, Sigma-Aldrich, St louis MO, USA) were used for serotonin hypersensitivity assay. All other reagents used were of the highest grade commercially available.

### Olfactory plasticity assay

Olfactory plasticity was analyzed as described previously [13,14]. The procedure is illustrated in Fig 1A. Briefly, CL2122 and CL2355 eggs were cultured on UV-NGM with or without

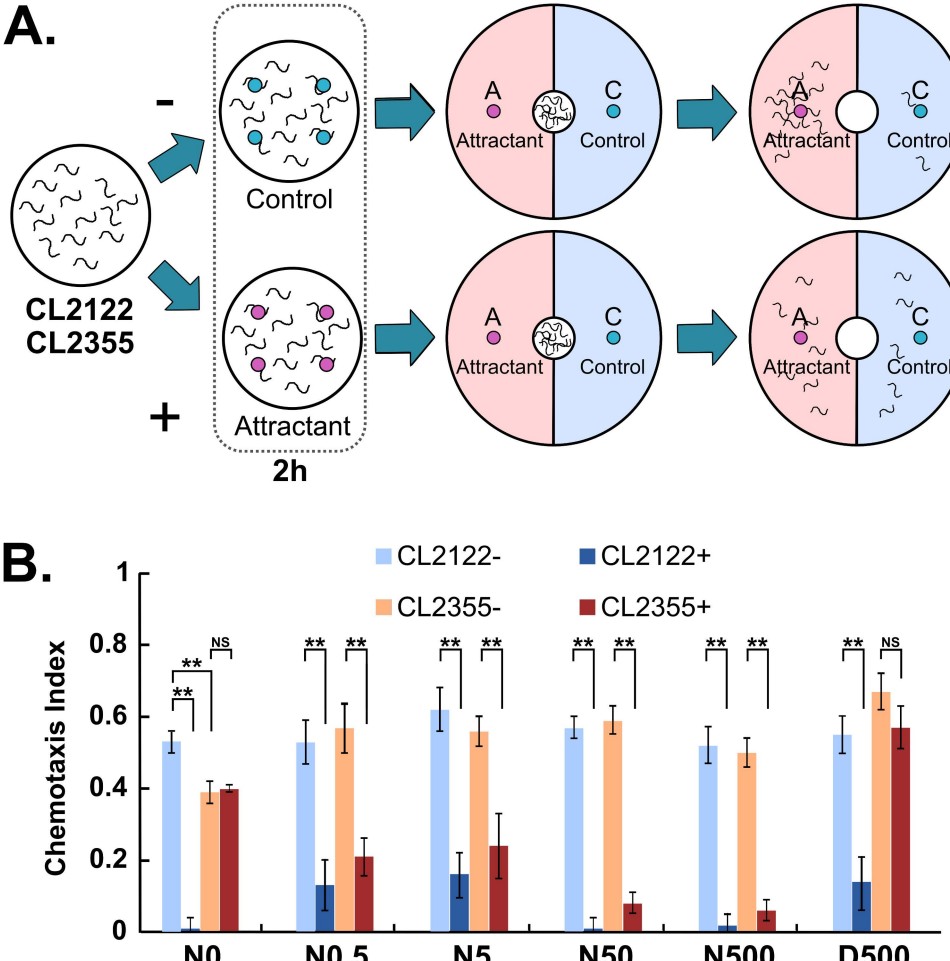

**Fig 1. Effect of Naturido on associative learning in the neuronal-A β transgenic strain. (A)** The schematic drawing illustrates the experimental procedures for olfactory plasticity assays. Worms were transferred to a plate without food, 48 h after the temperature upshift, and exposed to either 0.01% diacetyl (pre-exposed; [+]; red) or ethanol (diluent) (mock pre-exposed; [-];blue). They were then incubated for an additional 2 h before the assay. The configuration of the assay plate is shown in the center. The number of worms in regions A (pale pink) and **C** (pale blue) was counted to evaluate chemotaxis behavior (right). **(B)** Chemotaxis index for 0.01% diacetyl. CL2122 (no Aβ) and CL2355 (neuronal- Aβ) strain of transgenic worms were cultured in the absence (N0) or presence of Naturido (0.5 nM – 500 nM [N0.5 – N500]) or donepezil hydrochloride (500 nM [D500]) and then assayed. Statistical significance is shown only for the pairs mentioned in the text and for pairs focusing on the learning. **$p < 0.01$, [NS] Not significant. Error bars indicate SEM.

various concentrations of Naturido at 16°C for 48 h, then transferred to 23°C to induce $Aβ_{1-42}$ expression in CL2355. The worms were cultured at this temperature for another 48 h until they reached the young adult stage. Worms were then collected into 15 mL conical tubes by washing the culture plates with S-basal buffer (100 mM NaCl, 6 mM $K_2HPO_4$, 44 mM $KH_2PO_4$, 5 μg/mL cholesterol). They were washed twice with 10 mL of the buffer and once with sterilized water. For the training phase, worms were incubated either in the presence of 0.01% diacetyl (pre-exposed) or in the absence of the odorant (mock pre-exposed) on training plates (6 cm in diameter) containing assay agar (2% agar, 5 mM potassium phosphate buffer pH 6.0, 1 mM $CaCl_2$, 1 mM $MgSO_4$) for 2 h without food. In the pre-exposed condition, 2 μL of 0.01% diacetyl diluted in ethanol, were placed in the lid of the plate, which was then sealed with

parafilm (Fig 1A). After training, both pre-exposed and mock pre-exposed worms were separately collected into 15 mL conical tubes by washing the plates with S-basal buffer. The worms were washed with 10 mL of the buffer and leaved for 5 min at RT to allow worms to sediment at the bottom. Worms collected were then used for the chemotaxis assay.

In the chemotaxis assay, a 9-cm plastic dish containing 10 mL of assay agar was used. The configuration of the assay plate is shown in Fig 1A. To anesthetize worms that approached the odorant and control regions, 1 μL of 1 M $NaN_3$ was placed on points A and C, 2.5 cm apart from the center of the assay plate. Immediately after placing the $NaN_3$, approximately 30 worms were placed in the center of assay plate. Then, 1 μL of 0.01% diacetyl and 1 μL of ethanol (diluent) were placed on points A and C, respectively. The worms were allowed to move freely on the assay plate for 1 h, after which the number of worms found in regions A and C (regions shown in pale pink and pale blue, respectively, in Fig 1A) was counted. Worm behavior was evaluated using the chemotaxis index, defined as follows: (number of worms in region A - number of worms in region C)/ (total number of worms in region A and C). At least 10 plates were tested for each condition.

## Serotonin hypersensitivity assay

CL2122 and CL2355 eggs were cultured on UV-NGM with or without 50 nM Naturido at 16°C for 48 h. The plates were then transferred to 23°C and incubated for an additional 48 h. Worms were collected by washing the plates with M9 buffer (22 mM $KH_2PO_4$, 42 mM $Na_2HPO_4$, 86 mM NaCl, and 1 mM $MgSO_4$), and washed three times with the buffer. Subsequently, 50 worms were transferred to a 96-well assay plate (1 worm/well) containing 1 mM serotonin (creatinine sulfate salt) solution and left at RT for 5 min. Worms that did not exhibit body thrashing for 5 s were scored as paralyzed. Experiments were repeated 5 times for each condition.

## Locomotory activity assay

CL2122 and CL2355 eggs were cultured on UV-NGM with or without 50 nM Naturido at 16°C for 48 h. The plates were then transferred to 23°C and incubated for an additional 72 h. After incubation, the worms were transferred to a 6-cm dish containing 6 mL of assay agar. The number of body bends during 10 s of forward locomotion was counted under a dissecting microscope. At least 50 worms were tested for each condition.

## Lifespan analysis

Lifespan measurements were conducted as previously described [15]. CL2122, and CL2355 worms were cultured on UV-NGM with or without 50 nM Naturido at 16°C for 48 h. The worms were then transferred to new UV-NGM plates with or without Naturido, every two days and cultured at 23°C. Worms were considered dead if they failed to respond to a gentle touch by using a platinum wire worm pick. The number of surviving worms was recorded daily.

## Statistical analysis

All the experiments were repeated in five times or more, and the results are expressed as mean ± standard error of the mean (SEM). For multiple comparisons between control and transgenic strains, a one-way analysis of variance (ANOVA) was carried out, followed by post hoc testing using the Tukey-Kramer test with Satcel4 software (OMS, Saitama, Japan). A p-value of < 0.05 was considered statistically significant. Worm survival was analyzed using the Kaplan-Meier method, and significant difference between experimental treatments were assessed with the log-rank test by using JMP 17.2.0 software.

## Results

### Naturido alleviates the defect in associative learning caused by human Aβ$_{1-42}$ overproduction in neuronal cells of *C. elegans*

CL2355 is a transgenic *C. elegans* strain in which human Aβ$_{1-42}$ proteins are expressed in neuronal cells upon a temperature upshift from 16°C to 23°C. Probably due to the overexpression of Aβ$_{1-42}$ protein in their nervous system, worms of this strain exhibit several neurological phenotypes following the temperature upshift [3]. In this study, we have analyzed the associative learning of this neuronal-Aβ strain. We used diacetyl, a commonly used volatile chemoattractant, to assess *C. elegans* olfactory behavior. *C. elegans* exhibits a form of associative learning known as "olfactory plasticity" [13,14]. In this behavioral paradigm, well-fed young adult worms were strongly attracted by 0.01% diacetyl in a chemotaxis assay. However, when the worms were subjected to a certain period of starvation in the presence of diacetyl (pre-exposed: +, Fig 1A), their chemoattraction is diminished. This change in chemotaxis behavior is not observed when worms are subjected to starvation in the absence of diacetyl (mock pre-exposed: -, Fig 1A). Thus, the observed behavioral alteration is interpreted as a learned response resulting from the association between the odor and starvation in the *C. elegans* nervous system.

We conducted chemotaxis assays to assess the chemotaxis behavior of both the control strain (no Aβ) and the neuronal-Aβ strain, with or without preexposure to diacetyl under starvation conditions. In the absence of Naturido, the neuronal-Aβ strain showed slightly reduced chemotaxis behavior toward diacetyl compared to the no Aβ control in the mock pre-exposed condition. Furthermore, the behavior of the neuronal-Aβ strain after preexposure differed markedly from that of the no Aβ control, indicating that neuronal Aβ$_{1-42}$ overproduction impaired associative learning in *C. elegans*. In contrast, when the neuronal-Aβ strain was cultured in the presence of 0.5 nM or higher concentrations of Naturido, both the chemotaxis behavior and associative learning were significantly restored (Fig 1B). These results demonstrate that Naturido can alleviate the Aβ-induced adverse effects in the nervous system of *C. elegans*. Remarkably, donepezil hydrochloride (marketed as Aricept), a drug used to treat AD, did not restore associative learning significantly in the neuronal-Aβ strain when added to the culture plate. This suggests that Naturido may have slightly superior potency compared to donepezil hydrochloride in alleviating the impairment of associative learning caused by neuronal Aβ$_{1-42}$ overproduction in transgenic *C. elegans*.

### Naturido alleviates several other defects caused by human Aβ$_{1-42}$ overproduction in neuronal cells of *C. elegans*

Exogenous serotonin inhibits locomotion in wild-type *C. elegans*, leading to paralysis [16]. Presumably due to inefficient reuptake of serotonin by defective neurons at the relevant synapses, the neuronal-Aβ strain becomes hypersensitive to serotonin [3] (Fig 2A). We examined the effect of Naturido on serotonin hypersensitivity in the transgenic strain. The ratio of active worms was significantly increased in transgenic worms cultured in the presence of 50 nM Naturido, indicating that Naturido alleviated hypersensitivity to exogenous serotonin (Fig 2A, S1 Video). Possibly due to motor neuron dysfunction, the neuronal-Aβ strain shows a reduced locomotory activity on culture plate. In this experiment, both the transgenic and control worms cultured in the presence of 50 nM Naturido exhibited significant improvement on the locomotory activity (Fig 2B, S2 Video).

We further tested the effects of Naturido on the longevity of the transgenic strain with neuron-specific Aβ$_{1-42}$ overproduction. Notably, the addition of 50 nM Naturido had no effect on the longevity of control worms that did not express Aβ$_{1-42}$ (Fig 3A). The average lifespan of control worms was approximately 30 days, with a maximum lifespan of 40 days. Although

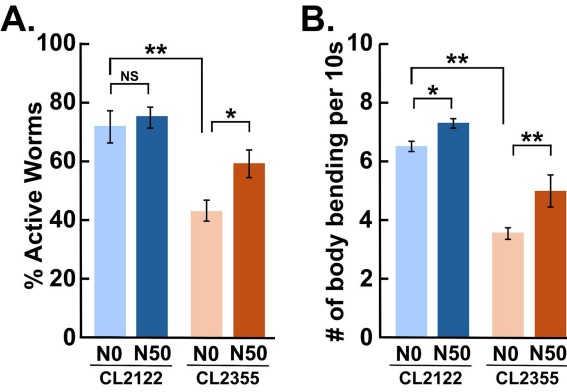

**Fig 2. Effect of Naturido on neuronal function in the neuronal-A β transgenic strain.** CL2122 (no Aβ) and CL2355 (neuronal-Aβ) worms were cultured in the absence (N0) or presence of Naturido (50 nM [N50]). Assays were performed 48 h after the temperature upshift. **(A)** Serotonin hypersensitivity assay. The ratio of active worms was scored. **(B)** Locomotory activity assay. The number of body bends per 10 sec was counted. Statistical significance is shown only for the pairs mentioned in the text and for the pairs focusing on the effect of Naturido. * $p < 0.05$, ** $p < 0.01$, NS Not significant. Error bars indicate SEM.

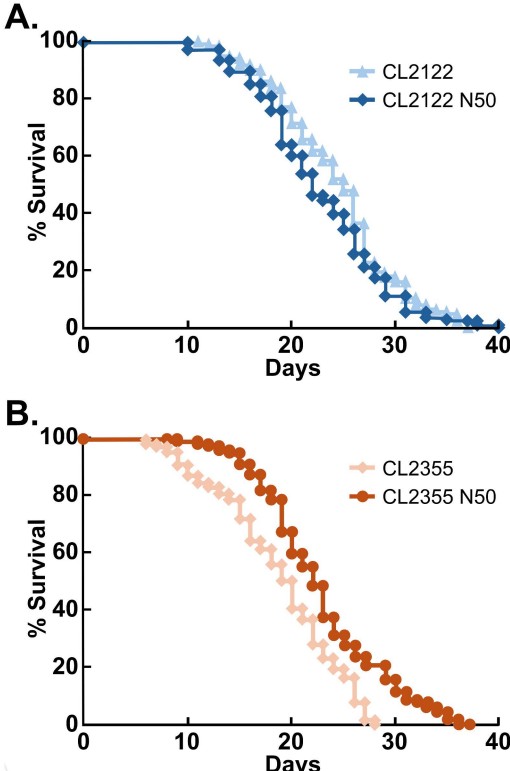

**Fig 3. Effect of Naturido on longevity in the neuronal-A β transgenic strain.** Longevity was assessed in control worms (CL2122, A) and neuronal-Aβ worms (CL2355, B), both cultured in the absence or presence of 50 nM Naturido (N50). The ratio of surviving worms after the temperature upshift is shown. Note that the longevity of the neuronal-Aβ strain was reduced compared to the control. The longevity of the neuronal-Aβ strain was restored by Naturido administration ($p < 0.01$, Kaplan–Meier analysis followed by log-rank test) At least 100 worms were tested for each condition.

lifespan can vary depending on environmental factors such as temperature and food quality, under our experimental conditions, control worms consistently exhibited a lifespan within this range. In contrast, addition of 50 nM Naturido significantly restored the shortened lifespan observed in the transgenic strain expressing $A\beta_{1-42}$ in neuronal cells ([Fig 3B]).

## Discussion

By using the transgenic *C. elegans* as a model animal, we demonstrated the potential of Naturido administration to alleviate adverse effects caused by overproduction of human $A\beta_{1-42}$ protein, including those related to Alzheimer's disease (AD), such as impaired associative learning. Naturido is a natural ingredient purified from the extract of the medicinal fungus (*Isaria japonica*) [8]. Fungal products have a long history of traditional medicine and generally considered safe [2,17,18]. Indeed, no detrimental effects were observed in control no-Aβ worms treated with Naturido in this study. Furthermore, our previous studies showed that oral administration of hot water extract of *I. japonica* or purified Naturido did not produce harmful effects in mouse model [8,9]. Therefore, no safety concerns have been associated with oral administrations of Naturido. One notable aspect of Naturido is its effective concentration. Improvements in associative learning were observed at a concentration of 0.5 nM, which is equivalent to 0.3 μg/kg of Naturido in water. In our previous studies, oral administration of hot water extract of *I. japonica* was conducted at 2.5 or 25 mg/kg/day [9], while purified Naturido was administered orally at 2.5 or 25 μg/kg/day to the senescence-accelerated mice [8]. Both treatments successfully improved the age-related deficits, including spatial learning and contextual learning, which are closely associated with dementia. Based on these results, the concentration of Naturido that alleviates Aβ-related symptoms in this study appears to be a practical dosage for its daily oral administration. Given its safety profile and effective dosage, Naturido emerges as an important candidate molecule for the treatment of AD.

## Supporting information

**S1 Video. A video of the serotonin hypersensitivity assay.** All experimental results presented in this study were evaluated using a single-worm assay format. Population assays were conducted solely to provide an overall perspective. Asterisks indicate inactive worms. The arrow indicates the position of a poorly visible worm.
(MP4)

**S2 Video. A video of the locomotory activity assay.**
(MP4)

## Acknowledgments

We thank the *Caenorhabditis* Genetics Center (University of Minnesota) for providing us with all the *C. elegans* strains used in this study. We also thank Ms. Kana Mochizuki and Ms. Sachi Sugawara for their technical assistance.

## Author contributions

**Conceptualization:** Koichi Suzuki.

**Formal analysis:** Piyamas Sillapakong, Tokumitsu Wakabayashi.

**Funding acquisition:** Koichi Suzuki.

**Investigation:** Piyamas Sillapakong, Tokumitsu Wakabayashi.

**Supervision:** Koichi Suzuki.

**Writing – original draft:** Tokumitsu Wakabayashi.

**Writing – review & editing:** Tokumitsu Wakabayashi.

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
