## [Decision Letter · Decision Letter 0]

23 Dec 2024

PONE-D-24-54123Naturido alleviates amyloid β_1-42_ -induced adverse effects in a transgenic *Caenorhabditis elegans* model of Alzheimer’s disease.PLOS ONE

Dear Dr. Wakabayashi,

Thank you for submitting your manuscript to PLOS ONE. After careful consideration, we feel that it has merit but does not fully meet PLOS ONE’s publication criteria as it currently stands. Therefore, we invite you to submit a revised version of the manuscript that addresses the points raised during the review process. 

The reviewer recommends that your manuscript be accepted following minor revisions.

We look forward to receiving your revised manuscript.

Kind regards,

David M. Ojcius

Academic Editor

PLOS ONE

Journal Requirements:

Reviewers' comments:

Reviewer's Responses to Questions

**Comments to the Author**

1. Is the manuscript technically sound, and do the data support the conclusions?

Reviewer #1: Yes

2. Has the statistical analysis been performed appropriately and rigorously? 

Reviewer #1: Yes

3. Have the authors made all data underlying the findings in their manuscript fully available?

Reviewer #1: Yes

4. Is the manuscript presented in an intelligible fashion and written in standard English?

Reviewer #1: Yes

5. Review Comments to the Author

Reviewer #1: The work " Naturido alleviates amyloid β 1-42 -induced adverse effects in a transgenic Caenorhabditis elegans model of Alzheimer’s disease", authored by Piyamas Sillapakong, Tokumitsu Wakabayashi, Koichi Suzuki,

demonstrates the effect of the cyclic peptide Naturido, extracted from the fungus Isaria Japanica, on a partial rescue from several age-related deficits in an AD model of C. elegans.

The paper focuses on the effects related to worms associative learning, serotonin hypersensitivity, locomotion issues and vitality.

Such effects are proven by considering the comparison of the AD model CL2355 with the control CL2122 worms that do not overexpress Ab1-42 under thermal switch from 16°C to 23°C.

I suggest some minor revisions before publication, in order to make some experiments more clear:

1) In the introduction the authors missed the citation to another work on an AD model of C.elegans involving the anti-AD effect of a natural protein, alpha-casein, due its idp and chaperone-like properties and its ability to sequester oligomeric intermediates in amyloid aggregation [Paterna et al. 2023, 10.1021/acschemneuro.3c00239].

2) In the description of Fig.1A, where the chemotaxix assay is described, is not clear if A and B are points or regions or even lines delimiting a circle of radius 2.5cm. Sorry, I could not make it to understand. Can you better explain?

3) What does HT assay means? I missed acronym list.

4) Comment on the fact that also mocks dye after 40 days. Is it the worm natural lifespan?

5) Comment on the fact that under the treatment with N50 it seems that a tiny toxicity is present in the CL2122 lifespan (not clear to me if it is statistically consistent?), while improvements are noticed in the locomotion and survival experiments.

6) Remember to give reference of all the products used, with brand and product number, in the right Reagents Section.

7) Some supplementary information, such as videos, could be helpful for the unexpert reader to have a representation of the main results of the study, like the rescue in body bending in the AD case for example?

6. PLOS authors have the option to publish the peer review history of their article (what does this mean? ). If published, this will include your full peer review and any attached files.

**Do you want your identity to be public for this peer review?** For information about this choice, including consent withdrawal, please see our Privacy Policy .

Reviewer #1: No

---

## [Author Response · Author response to Decision Letter 1]

18 Feb 2025

Dear Reviewer #1,

Thank you for your review comments on our manuscript (PONE-D-24-54123). Before responding point-by point, we would like to express our deep thanks for your constructive and thoughtful comments. We have taken these comments and suggestions into account in our revised manuscript as described below.

We hope that the revised manuscript is now suitable for publication in PLOS ONE.

RE: Reviewer #1

1) In the introduction the authors missed the citation to another work on an AD model of C.elegans involving the anti-AD effect of a natural protein, alpha-casein, due its idp and chaperone-like properties and its ability to sequester oligomeric intermediates in amyloid aggregation [Paterna et al. 2023, 10.1021/acschemneuro.3c00239].

- Thank you for this suggestion. We are sorry that we missed this important study. According to this comment, we cited this study in the “Introduction” section.

2) In the description of Fig.1A, where the chemotaxix assay is described, is not clear if A and B are points or regions or even lines delimiting a circle of radius 2.5cm. Sorry, I could not make it to understand. Can you better explain?

- We are very sorry that we did not describe this point explicitly. We revised Fig. 1 by highlighting the regions used for counting. Additionally, we renamed the region B

to region C (for control), and revised the manuscript accordingly.

3) What does HT assay means? I missed acronym list.

- We apologize for the lack of consistency in terminology. The HT assay refers to the 5-hydoroxytryptamine (serotonin) assay. In the revised manuscript, we consistently used the term “serotonin hypersensitivity assay” throughout the text.

4) Comment on the fact that also mocks dye after 40 days. Is it the worm natural lifespan?

- Thank you for your constructive comments. The average lifespan of the control worms was approximately 30 days, with a maximum of 40 days. Although lifespan depends on environmental factors such as temperature and food quality, under our experimental conditions, the control worms consistently exhibited a lifespan within this range. We have added the description to the “Results” section.

5) Comment on the fact that under the treatment with N50 it seems that a tiny toxicity is present in the CL2122 lifespan (not clear to me if it is statistically consistent?), while improvements are noticed in the locomotion and survival experiments.

- According to this comment, we performed a Wilcoxon test in addition to the log-rank test for the lifespan data, but neither test revealed statistically significant differences between CL2122 and CL2122 N50 results. We also conducted a comprehensive reassessment of the statistical analyses for the entire study (chemotaxis, serotonin hypersensitivity, and locomotion) and revised some descriptions and figures. Although no significant differences were observed in the serotonin hypersensitivity of the control worms, significant differences were observed in locomotion, and this has been noted in the revised text.

6) Remember to give reference of all the products used, with brand and product number, in the right Reagents Section.

- We are very sorry that we did not describe this point explicitly. We have added the product number of donepezil hydrochloride, as well as the brand and product number of serotonin. For other reagents, we used commonly available products, and this point was described in the revised manuscript.

7) Some supplementary information, such as videos, could be helpful for the unexpert reader to have a representation of the main results of the study, like the rescue in body bending in the AD case for example?

- Thank you for this suggestion. We recorded videos of several experiments and included them as supplementary materials in our revised manuscript.

---

## [Decision Letter · Decision Letter 1]

23 Feb 2025

Naturido alleviates amyloid β_1-42_ -induced adverse effects in a transgenic *Caenorhabditis elegans* model of Alzheimer’s disease.

PONE-D-24-54123R1

Dear Dr. Wakabayashi,

We’re pleased to inform you that your manuscript has been judged scientifically suitable for publication and will be formally accepted for publication once it meets all outstanding technical requirements.

Kind regards,

David M. Ojcius

Academic Editor

PLOS ONE

Additional Editor Comments (optional):

Reviewers' comments:

Reviewer's Responses to Questions

**Comments to the Author**

1. If the authors have adequately addressed your comments raised in a previous round of review and you feel that this manuscript is now acceptable for publication, you may indicate that here to bypass the “Comments to the Author” section, enter your conflict of interest statement in the “Confidential to Editor” section, and submit your "Accept" recommendation.

Reviewer #1: All comments have been addressed

2. Is the manuscript technically sound, and do the data support the conclusions?

Reviewer #1: Yes

3. Has the statistical analysis been performed appropriately and rigorously? 

Reviewer #1: Yes

4. Have the authors made all data underlying the findings in their manuscript fully available?

Reviewer #1: Yes

5. Is the manuscript presented in an intelligible fashion and written in standard English?

Reviewer #1: Yes

6. Review Comments to the Author

Reviewer #1: (No Response)

7. PLOS authors have the option to publish the peer review history of their article (what does this mean? ). If published, this will include your full peer review and any attached files.

**Do you want your identity to be public for this peer review?** For information about this choice, including consent withdrawal, please see our Privacy Policy .

Reviewer #1: No

---

## [Editor Report · Acceptance letter]

PONE-D-24-54123R1

PLOS ONE

Dear Dr. Wakabayashi,

I'm pleased to inform you that your manuscript has been deemed suitable for publication in PLOS ONE. Congratulations! Your manuscript is now being handed over to our production team.

Kind regards,

on behalf of

Dr. David M. Ojcius

Academic Editor

PLOS ONE